# Temperature Difference in Loading Area (Tarmac) during Handling of Air Freight Operations and Distance of Production Area Affects Quality of Fresh Mango Fruits (*Mangifera indica* L. 'Nam Dok Mai Si Thong')

Kraisuwit Srisawat [1,*], Panmanas Sirisomboon [2], Umed Kumar Pun [2], Warawut Krusong [3], Samak Rakmae [1], Nattawut Chaomuang [1], Pornkanya Mawilai [1], Thadchapong Pongsuttiyakorn [1], Chalisa Chookaew [1] and Pimpen Pornchaloempong [1,*]

[1] Department of Food Engineering, School of Engineering, King Mongkut's Institute of Technology Ladkrabang, Bangkok 10520, Thailand
[2] Department of Agricultural Engineering, School of Engineering, King Mongkut's Institute of Technology Ladkrabang, Bangkok 10520, Thailand
[3] Division of Fermentation Technology, School of Food Industry, King Mongkut's Institute of Technology Ladkrabang, Bangkok 10520, Thailand
* Correspondence: kraisuwit@gmail.com (K.S.); pimpen.po@kmitl.ac.th (P.P.)

**Abstract:** Mango (*Mangifera indica* L.) 'Nam Dok Mai Si Thong' is an important cultivar for export from Thailand. Export mainly takes place via air transport, but for about 2 h at the loading area (tarmac), unit loading devices (ULDs) are exposed to ambient environmental conditions. In this research, the effects of different temperature conditions at the loading area (tarmac) and the distance of the production area from the tarmac on the quality of fresh mango fruits were studied. The treatments included three temperature conditions for 2 h (simulated handling in tarmac)—constant temperature (20 °C), non-insulated or insulated and exposed to sun—and two distances of the tarmac from the production area—short distance (i.e., transport occurring 53 h after harvest) and long distance (i.e., transport occurring 70 h after harvest). The temperature variation in the boxes exposed to the sun was greater in the non-insulated than in the insulated boxes, but this effect was more pronounced in fruit from the short-distance production area (28.1 °C insulated and 36.9 °C non-insulated) than in fruit from the long-distance production area (34.2 °C insulated and 38 °C non-insulated). Insulation and short distance increased the shelf life, decreased weight loss, delayed the decrease in average firmness and rupture force, etc. The insulation of mango fruit boxes mitigates the deleterious effect of exposure to 2 h of direct sun by reducing the increase in temperature, thus improving the shelf life and quality of mango fruit.

**Keywords:** *Mangifera indica*; Nam Dok Mai Si Thong; air freight; postharvest quality; quality attributes

## 1. Introduction

Mango (*Mangifera indica* L.) is a climacteric fruit and is a popular tropical fruit [1]. It is the sixth most widely grown fruit crop in the world, and Thailand ranked as the tenth highest producer in 2019–2020 [2–4]. In 2021, the total area under mango cultivation in Thailand was 146,217 hectares, with a fruit yield of more than 905,934 tons [5]. Thailand exports of fresh mangoes were valued at USD 734 million in 2020 [6]. Thailand has many mango varieties, and more than 60 types are commercially cultivated [7]. However, one of the most important leading mango cultivars for export is 'Nam Dok Mai Si Thong' [8]. The skin of the 'Nam Dok Mai Si Thong' mango is yellow, and it turns to golden-yellow with ripening. This variety is sweet, fragrant, and juicy, and has no fibrous tissue [9].

Consumer acceptance of mangoes is correlated with the internal and external quality factors of mango fruit [10]. The quality of mango is difficult to maintain, because mango

is a highly perishable fruit that ripens rapidly after harvest [3]. Hence, mango fruits have a short storage life after harvest due to poor postharvest handling, temperature, disease incidence, and sensitivity to chilling injury [11,12]. Quality losses in fresh fruit or rejection of the whole loads can result from temperature abuses encountered during handling [13]. Air freight is the fastest way to transport highly perishable horticultural crops over long distances [14]. However, normal air transport involves a significant break in the cold chain of perishable handling. The major causes for this rupture are the fluctuating temperatures that often occur during air freight and ground operations [15].

A schematic diagram of mango fruit transportation from the farm through the airport to the customer is shown in Figure 1. Studies measuring time–temperature profiles during the air freight of perishable foods indicate improper control of the temperature. It is estimated that only approximately half of the transit time attributed to air freight is spent in flight; during the other half of the time, the perishable food is being transported to or from the airport, stored at the airport, or loaded into or unloaded from the plane on the tarmac [16]. High temperatures can occur inside the unit loading devices (ULDs), especially when the perishable food is on the tarmac before being loaded into the airplane: on the tarmac, temperatures as extreme as $-50\,°C$ or $+50\,°C$ can be observed, depending on the location or season [13,17,18]. Furthermore, the studies of Villeneuve et al. [19] report the occurrence of a difference of about $14\,°C$ inside ULDs between the worst-case and best-case scenarios. Temperature variations inside ULDs during airport operations can be caused by the presence of cooling and handling equipment at the airport, the type of ULDs used for the transportation of perishables (plastic, metal, or insulated), flight delays, solar radiation on the tarmac, and the airport's location [19–21].

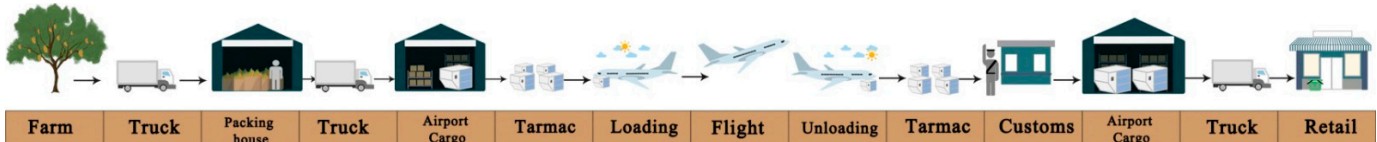

**Figure 1.** Flow diagram of transportation of mango fruits through the airport from the farm to the customers.

Difficulties in controlling temperature during normal handling procedures such as loading, unloading, air freight or truck transportation, warehouse storage and retail have an impact on fruit and vegetables, which often encounter inappropriate temperatures. Observations at Miami airport have shown that containers with fresh produce are held for a minimum of 2 h at temperatures of $30\,°C$ or higher before being loaded into the airplane [13]. In addition, fluctuating temperatures during the distribution of fresh fruit and vegetables may negatively affect their quality. For example, in a previous study, Nunes and Emond showed that strawberries stored in fluctuating temperatures exhibited more weight loss, higher pH, less firmness, and lower glucose content than those stored at constant temperature, even though the degree totals each day were the same [22].

Any fresh produce's postharvest supply chain must include transportation as a critical step [23]. Additionally, lengthy transportation times during the shipping of fresh produce can hasten enzymatic and metabolic processes, increasing the danger of mechanical damage and lowering market value [24]. For example, a previous study by Al-Dairi and others [25] showed that different transportation distances significantly affected tomato physical quality parameters. This study evaluated physical changes in tomatoes during transportation and storage. Tomatoes were transported over three distances (100, 154, and 205 km) from a local farm and delivered to the Postharvest Laboratory. The results showed that long transportation distances greatly increased the weight and firmness reduction, and produced greater colour changes during storage.

Therefore, the main purpose of this research is to evaluate the effects of different temperatures in loading areas (tarmac) different distances from the mango production

areas on the quality of the mango with respect to weight loss, firmness, total soluble solids, acidity, pH, colour change and visual assessment.

## 2. Materials and Methods

### 2.1. Materials

Mango cv. 'Nam Dok Mai Si Thong' from Chachoengsao province (13.606977° N, 101.298767° E) in the eastern part of Thailand for the first harvest (short distance) and Khon Kaen province (16.14964° N, 102.76750° E) in the northeastern part of Thailand for the second harvest (long distance) were used for this study. Mangoes of uniform shape and size (350–450 g) and at commercial harvesting stage, absent of defects due to cracks and diseases, were selected according to the Thai Agriculture Standard for export mango [26]. Chachoengsao is one of the five provinces in which mangoes are planted the most in Thailand, as the soil in this area is suitable for planting mangoes [27]. Furthermore, the province of Khon Kaen counts among the major producers of the 'Nam Dok Mai Si Thong' mango, with around 10% of the total export coming from Khon Kaen [28].

### 2.2. Mango Sample Preparation

Mango fruits of exportable grade were picked from an orchard in Chachoengsao, Thailand, on 8 March 2022 (first harvest) and picked from an orchard in Khon Kaen, Thailand on 21 June 2022 (second harvest). Mangoes were collected, graded, packed, and transported to the packing house (13.80253 °N, 100.06714 °E) in a regular pickup truck. Therefore, fruits from the first harvest were picked and transported within 27 h after treatment at the packing house, while those from the second harvest were picked and transported within 44 h after treatment at the packing house. The fruits were cleaned up and exposed to hot water treatments to kill fruit flies at the packing house in accordance with EU mango import regulations [29]. After drying, the fruits were placed in a foam net bag and packed in a 5-layer corrugated cardboard box for air transportation. The carton size was 40 × 52 × 10 cm, with 2 side vents on each side of the box with a size of 7 × 2.5 cm, and closed with diagonal net not exceeding 1.6 mm. The weight of fruits per carton was approximately 5 kg, and the time of processing in packing house was 9 h. After this, fruits were transported to the Food and Smart Agriculture Engineering KMITL laboratory within 2 h and stored at 20 °C for 15 h before the simulation of handling treatment. Therefore, the age of fruits after harvest in the first harvest and the second harvest after simulating handling treatment was 53 and 70 h, respectively. A constant temperature of 20 °C was used as a reference temperature, since mangoes matured at a temperature of 20–23 °C demonstrate good appearance and eating quality [30].

### 2.3. Simulated Handling Treatments

The fruits were grouped into 3 treatments. A total of 10 cartons per treatment and an arrangement of 2 boxes per shelf of 5 layers covered with Low-Density Polyethylene (LDPE) plastic were separated into three sections with treatments consisting of: (1) placed in a 20 °C temperature room throughout the shelf life; (2) placed outdoors; and (3) placed outdoors covered with polystyrene foam insulation with a thickness of 2.54 cm. The second and third treatments were placed outdoors for 2 h to simulate transport conditions at the tarmac. King Mongkut's Institute of Technology Ladkrabang (KMITL) is not far from Suvarnabhumi Airport (about 5 km). Therefore, the temperature and relative humidity were assumed not to be significantly different. After outdoor treatment for 2 h, the fruits were returned to constant room temperature at 20 °C as the control group. The polystyrene foam insulation was removed from the third treatment, and LDPE plastic was removed from all treatments after 16 h to simulate transport conditions on the airplane and at the tarmac. The quality assessment of fruits was performed prior to the simulation at the tarmac, after simulation at the tarmac, after transportation by airplane, and throughout the shelf life using ten fruits per treatment at all steps, resulting in a total of one hundred fruits per treatment, as shown in Table 1. Handling times and temperature during simulated

handling from the tarmac to the consumer are shown in Table 1, and were chosen as being representative for the purposes of our measurements (unpublished) on the basis of the required air freight time to transport mango fruits from Suvarnabhumi Airport, Thailand to Paris Charles de Gaulle Airport, France.

**Table 1.** Time and temperature used for 'Nam Dok Mai Si Thong' mangoes during simulation of handling from the tarmac to the customer under constant, non-insulated, and insulated conditions.

| Handling Simulation | Time | | Temperature (°C)) | | |
|---|---|---|---|---|---|
| | (h) | Cumulative (h) | First Treatment Constant | Second Treatment Non-Insulated | Third Treatment Insulated |
| Warehouse | 0 | 0 | 20 [a] | 20 | 20 |
| Tarmac | 2 | 2 | 20 | Real Temp [b] | Real Temp [b] |
| Airplane | 12 | 14 | 20 | 20 | 20 |
| Tarmac | 2 | 16 | 20 | 20 | 20 |
| Truck + Store | 10 | 26 | 20 | 20 | 20 |
| Shelf life (2 day) | 48 | 74 | 20 | 20 | 20 |
| Shelf life (4 day) | 48 | 122 | 20 | 20 | 20 |
| Shelf life (6 day) | 48 | 170 | 20 | 20 | 20 |
| Shelf life (8 day) | 48 | 218 | 20 | 20 | 20 |
| Shelf life (10 day) | 48 | 266 | 20 | 20 | 20 |

[a] $20.0 \pm 1$ °C and 90.5% RH. [b] $40.0 \pm 2$ °C and 70% RH.

### 2.4. Temperature and Relative Humidity Monitoring

The temperatures inside and outside of the corrugated cardboard box of mango fruits were monitored throughout the storage using temperature data loggers (iButtonDS1922L-F5, Maxim Integrated Products, Inc., San Jose, CA, USA). Relative humidity (RH) was monitored outside the corrugated cardboard box with a mini temperature and humidity data logger (Testo 174H, Testo SE & Co., Titisee-Neustadt, Germany).

### 2.5. Shelf Life

The shelf life of fruit was visually assessed on the basis of the colour, shrivelling and decay 53 h after harvest in H1 (short distance) and 70 h after harvest in H2 (long distance). Assessment was performed at 48 h intervals until the fruit was deemed marketable.

### 2.6. Weight Loss

The weight loss of the mango fruit samples was calculated as per Shah and Hashmi [31]. The loss of weight was calculated from the initial weight of ten individual fruits per treatment. The weight loss of the fruits was determined by weighing at the beginning of the experiment and during storage. The same was expressed as the percentage loss of the initial weight (Equation (1)).

$$\text{Weight loss (\%)} = [(W1 - W2)/W1] \times 100 \tag{1}$$

W1 is the weight of initial fruit.
W2 is the weight at the time of sampling.

### 2.7. Firmness

The initial firmness, average firmness, rupture force and toughness of the mango fruit samples were measured using a texture analyser (TA-HD. Plus, Stable Micro Systems, Surrey, UK) equipped with a stainless-steel cylinder probe 5 mm in diameter. The pre-test, test, and post-test speeds were 1, 0.2, and 10 mm s$^{-1}$, respectively. Measurements were performed on opposite sides of each fruit at the centre for a total of two measurements per fruit. The average of the two measurements was used, and the test was performed in three fruits per treatment.

2.7.1. Non-Destructive

Firmness of mango fruit samples was measured as per Penchaiya et al. [32] with some modification. A non-destructive limited force compression at 1 Newton (N) compression was performed. Mango fruits with peel were placed in a box of dry sand on the platform of the texture analysers. All fruits were measured on both sides, and the values obtained on both were recorded, but the averaged value was used to represent fruit firmness as a whole.

2.7.2. Destructive

The firmness of mango fruit samples was measured as per Sirisomboon and Pornchaloempong [33]. A destructive penetration test of the flesh was performed after removing the skin of the fruit (approximately 0.5 cm thick). The flesh was penetrated using the probe at the same position of the non-destructive compression measurement. The obtained force–distance curves were further analysed using Exponent Stable Micro System software (version 6.1.9, Stable Micro Systems, Surrey, UK) to indicate the initial firmness, average firmness, toughness, and rupture force in the flesh.

*2.8. Total Soluble Solids (TSS), Acidity and pH*

The total soluble solids and acidity (citric acid) of the mango fruit samples were measured as per Xu et al. [34]. The TSS and acidity of the resulting clear juice samples were measured using a pocket Brix-Acidity Meter (ATAGO Model PAL-BX/ACID1, Tokyo, Japan). After the peel and seeds were removed, the centre of the flesh of both the sides were scraped using a fine grater, homogenized, and filtered through tea filter bag. One or two drops of fruit juice were placed on the Brix-Acidity meter to measure the TSS directly. To measure the acidity, the juice was diluted with distilled water in a ratio of 1:50, then one or two drops of the diluted juice were used for measurement. Each sample was measured three times, and the TSS and acidity for that sample were recorded as the average of these three values. Between each measurement, the Brix-Acidity meter was calibrated with distilled water. The unit of acidity is $gL^{-1}$.

The pH of the juice samples was determined using a digital pH meter (OHAUS ST300, NJ, USA) previously standardized to pH 4 and pH 7 following the standard method of AOAC [35].

The TSS, acidity and pH of fruit samples were measured on opposite sides of each fruit at the centre for a total of two measurements per fruit. Three replicate fruits were measured, and the average was calculated.

*2.9. Colour Measurement*

The skin and flesh colour near the seed of the mango fruit samples was measured as per Medina-Rendon et al. [36] using a spectrophotometer (model MiniScan EZ 4500 L (Hunter Associates Laboratory Inc., VA, USA)). Colour was measured at the top (near the stem), centre, and bottom of each fruit at both sides. The obtained values were averaged to represent the colour of each fruit. The colour was recorded using the CIE–$L^*$ $a^*$ $b^*$ uniform colour space, where '$L^*$' indicates lightness, '$a^*$' indicates chromaticity on the green ($-$) to red ($+$) axis, and '$b^*$' indicates chromaticity on the blue ($-$) to yellow ($+$) axis.

Yellowness Index (YI) was calculated as defined by the formula of Francis and Clydesdale (Equation (2)) [37]:

$$YI = 142.86 b^*/L^* \qquad (2)$$

*2.10. Visual Quality*

2.10.1. Visual Colour

The colour of each individual mango fruit was assessed using a visual rating scale as per Jacobi et al. [38], where 1 = very light yellow, 2 = light yellow, 3 = mostly yellow, 4 = predominantly yellow, and 5 = fully orange-yellow, as shown in Figure 2a. The peel colour was not considered to be a limitation to acceptability for sale.

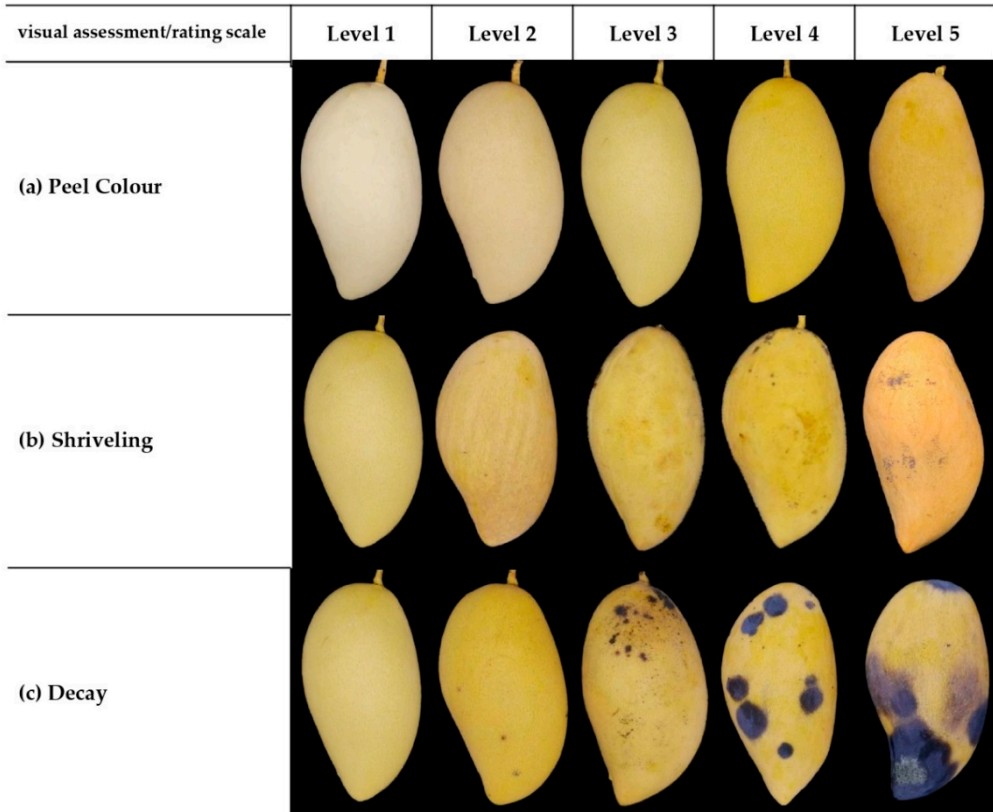

**Figure 2.** (**a**) Peel colour rating scale, (**b**) shrivelling rating scale, and (**c**) decay rating scale of 'Nam Dok Mai Si thong' mango fruits.

### 2.10.2. Shrivelling

The shrivelling of each individual mango fruit was assessed using a visual rating scale as per Quintana and Paull [39], where 1 = none, field-fresh, no signs of shrivelling, 2 = slight, minor signs of shrivelling, not objectionable, 3 = moderate, shrivelling evident, becoming objectionable, 4 = severe shrivelling, definitely objectionable, and 5 = extremely shrivelled, wrinkled, and dry, not acceptable under normal conditions, yellow, as shown in Figure 2b. A shrivelling rating of 3 was considered to be the limit of acceptability for sale.

### 2.10.3. Decay

The decay of each individual mango fruit was assessed using a modified visual rating scale as per Nunes et al. [40], where 1 = 0%, no decay, 2 = 1–25% decay, probable decay (brownish/grayish sunken minor spots), 3 = 26–50% decay, slight to moderate decay (spots with decay and some mycelium growth), 4 = 51–75% decay, moderate to severe decay, 5 = 76–100% decay, severe to extreme decay (the mango is either partially or completely rotten), as shown in Figure 2c. A decay rating of 3 was considered to be the limit of acceptability for sale.

### 2.11. Statistical Analysis

The experimental setup employed a completely randomized design (CRD). The statistical analysis of data was performed by one-way analysis of variance (one-way ANOVA), and the differences between means were analysed with Duncan's test ($p < 0.05$) using the SPSS Statistic Version 28.0.0.0(190) software (IBM Corporation, New York, NY, USA).

### 3. Results

*3.1. Temperature and Relative Humidity Monitoring*

High temperature and low relative humidity are critical factors that negatively influence the postharvest shelf life of mango fruits. The increase in temperature in the boxes containing mango fruits was influenced by both the distance of production area from the tarmac and the difference in packaging. Temperature variation in the boxes exposed to the sun was higher in the non-insulated than in the insulated case, but effectiveness was better in the short-distance (28.1 °C insulated and 36.9 °C non-insulated) than in the long-distance (34.2 °C insulated and 38 °C non-insulated) case. Fruits transported from a production area a long distance away experienced higher temperature in the boxes irrespective of the packaging and outside temperature. Similarly, while packaging was effective for reducing the temperature of the boxes containing the mango fruits, it was more effective when the fruits were transported over short distances.

From the first harvest (short-distance production area), the temperature during the handling simulation on the tarmac under the three different conditions is shown in Figure 3a. The temperature in the room for the constant temperature condition was 20.7 ± 0.4 °C, while the outdoor temperature was approximately 42.7 ± 1.9 °C. After the placement of the box of 'Nam Dok Mai Si Thong' Mango outdoors for 120 min, the temperatures inside the corrugated cardboard boxes in the (constant), non-insulated and insulated cases were (21.3 ± 0.2 °C), 36.9 ± 3.2 °C and 28.1 ± 0.6 °C, respectively. Additionally, the control room's relative humidity was 91.9 ± 2.0%, whereas outdoors it was 77.8 ± 0.7%, as shown in Table 2.

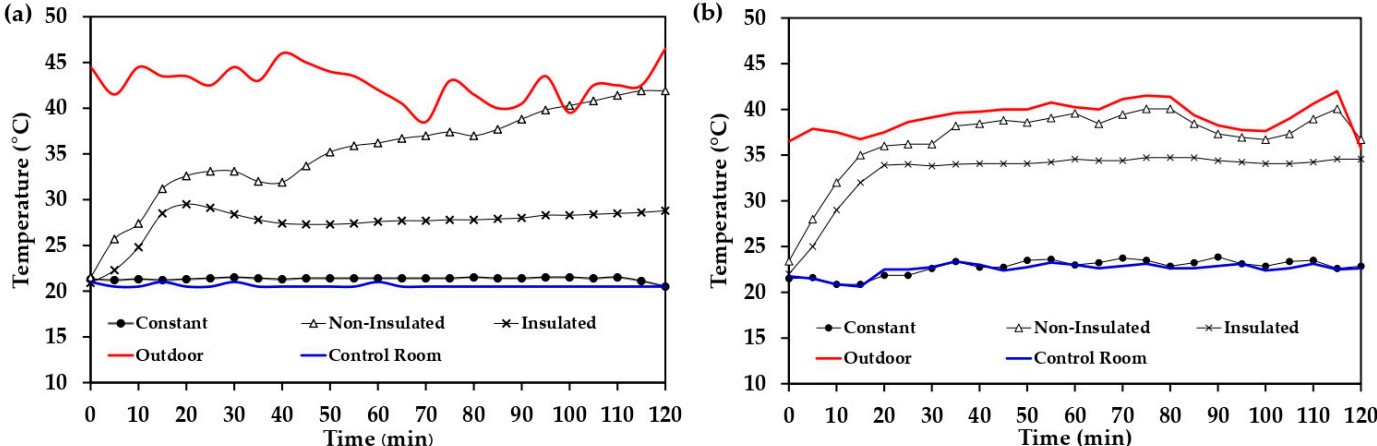

**Figure 3.** (**a**) First harvest temperature in the box, (**b**) second harvest temperature in boxes of mango fruits ('Nam Dok Mai Si thong') under constant, non-insulated, and insulated conditions during simulated handling at the tarmac.

**Table 2.** Temperature (°C) and relative humidity (%) during the handling simulation on the tarmac.

| Production Area | Type of Packaging | Treatment at the Tarmac | Temperature (°C) | Relative Humidity (%) |
|---|---|---|---|---|
| First Harvest Short Distance (53 h *) | LDPE LDPE LDPE and insulation | Outdoor Control room Constant temperature 20 °C Under sun Under sun | 42.7 ± 1.9 20.7 ± 0.4 21.3 ± 0.2 36.9 ± 3.2 28.1 ± 0.6 | 77.8 ± 0.7 91.9 ± 2.0 |
| Second Harvest Long Distance (70 h *) | LDPE LDPE LDPE and insulation | Outdoor Control room Constant temperature 20 °C Under sun Under sun | 39.1 ± 1.7 19.8 ± 0.5 22.8 ± 0.9 38.0 ± 1.3 34.2 ± 0.4 | 69.7 ± 3.6 90.1 ± 3.8 |

* Hours after harvest from the tree to KMITL.

From the second harvest (production area located a long distance away), the temperature during the handling simulation on the tarmac under three conditions is shown in Figure 3b. The temperature in the room for the constant temperature condition was 19.8 ± 0.5 °C, while the outdoor temperature was approximately 39.1 ± 1.7 °C. After placement of the box of 'Nam Dok Mai Si Thong' mango outdoors for 120 min, the temperatures inside the corrugated cardboard boxes in the (constant), non-insulated and insulated cases were (22.8 ± 0.9 °C), 38.0 ± 1.3 °C and 34.2 ± 0.4 °C, respectively. Additionally, the control room's relative humidity was 90.1 ± 3.8%, whereas the relative humidity outside was 69.7 ± 3.6%, as shown in Table 2. The temperature inside the box under polystyrene foam insulation of the second harvest increased to about 34 °C during the first 20 min, and the temperature was relatively stable until completion of 120 min.

*3.2. Shelf Life*

Shelf life is one of the most important postharvest parameters of mango fruit for better market reach, especially for export. The shelf life of Mango fruit was greater in the first harvest (short distance) by 48 h than in the second harvest (long distance), as shown in Table 3. This difference is mainly because of the longer time after harvest for the second harvest (70 h) compared to the first harvest (53 h). Similarly, LDPE and constant temperature (20 °C) and LDPE and insulation increased the shelf life by 48 h (2 days) compared to LDPE without insulation. The effectiveness of LDPE and constant temperature (20 °C) and LDPE and insulation is irrespective of whether the distance is short or long (i.e., belonging to the first or second harvest), and increases shelf life by 48 h in either case. However, the benefit is much higher when the mango production area is close to the airport.

**Table 3.** Shelf life of mango fruits after simulated handling treatments.

| Production Area | Type of Packaging | Treatment at the Tarmac | Shelf Life | |
|---|---|---|---|---|
| | | | Hours | Days |
| First Harvest Short Distance (53 h *) | LDPE | Constant temperature 20 °C | 170 | ≈7 |
| | LDPE | Under sun | 122 | ≈5 |
| | LDPE and insulation | Under sun | 170 | ≈7 |
| Second Harvest Long Distance (70 h *) | LDPE | Constant temperature 20 °C | 122 | ≈5 |
| | LDPE | Under sun | 74 | ≈3 |
| | LDPE and insulation | Under sun | 122 | ≈5 |

* Hours after harvest from the tree to KMITL.

*3.3. Quality Assessment*

3.3.1. Weight Loss

Weight loss is associated with the deterioration of fruit quality due to transpiration and other metabolic activities. Weight loss percentage increased with storage time in all fruits, irrespective of treatment, as shown in Table 4. Weight loss percentage was slightly greater in fruits from the long-distance production area (second harvest) compared to in those from the short-distance production area (first harvest), irrespective of the different tarmac treatments. The slightly greater weight loss in fruits from the long-distance production area could be due to the fact that the greater amount of time following harvest (70 h after harvest) results in a higher temperature built up in the simulated boxes. The constant temperature (20 °C) slowed down weight loss much more effectively than the insulated box or non-insulated boxes exposed to the sun, but was more effective in the long-distance production area (H2) than the short-distance production area (H1).

**Table 4.** Weight loss (%) of mango fruits after simulated handling treatments in first harvest.

| Treatment/Hour | Weight Loss (%) | | | | | | |
|---|---|---|---|---|---|---|---|
| | 2 | 26 | 74 | 122 | 170 | 218 | 266 |
| **H1** | | | | | | | |
| Constant | 0.08 ± 0.02 [c] | 0.72 ± 0.17 [c] | 2.34 ± 0.31 [b] | 4.04 ± 0.44 [a] | 6.17 ± 0.56 [a] | 7.48 ± 0.77 [a] | 9.38 ± 0.98 [b] |
| Non-Insulated | 0.15 ± 0.02 [a] | 1.35 ± 0.22 [a] | 2.74 ± 0.33 [a] | 4.34 ± 0.48 [a] | 6.32 ± 0.67 [a] | 8.03 ± 0.83 [a] | 10.08 ± 1.47 [a] |
| Insulated | 0.10 ± 0.02 [b] | 0.93 ± 0.16 [b] | 2.53 ± 0.37 [ab] | 4.16 ± 0.56 [a] | 6.23 ± 0.75 [a] | 7.83 ± 0.89 [a] | 9.71 ± 0.55 [ab] |
| **H2** | | | | | | | |
| Constant | 0.10 ± 0.02 [b] | 1.14 ± 0.14 [b] | 2.86 ± 0.25 [b] | 4.50 ± 0.35 [b] | 6.10 ± 0.46 [b] | 7.90 ± 0.56 [a] | 9.49 ± 0.65 [a] |
| Non-Insulated | 0.18 ± 0.04 [a] | 1.28 ± 0.13 [a] | 3.23 ± 0.14 [a] | 5.05 ± 0.22 [a] | 6.96 ± 0.82 [a] | 8.36 ± 0.33 [a] | 10.03 ± 0.43 [a] |
| Insulated | 0.13 ± 0.03 [b] | 1.18 ± 0.10 [ab] | 3.13 ± 0.29 [a] | 4.74 ± 0.42 [b] | 6.62 ± 0.46 [ab] | 8.20 ± 0.51 [ab] | 9.66 ± 0.59 [a] |

Data from the two harvests were analyzed separately. H1, first harvest; H2, second harvest; All values indicate mean ± S.D. ($n = 10$), and values with various superscript alphabetic letters indicate significant differences in each column within each harvest according to one-way analysis of variance (ANOVA) with Duncan's test ($p < 0.05$).

In H2, the constant temperature significantly reduced weight loss until 170 h as compared with the insulated or non-insulated treatments exposed to the sun. However, in H1, the constant temperature only significantly reduced weight loss compared to the other treatments until 26 h, and was on par with the insulated box by 74 h. It can be observed that in H1, weight loss stabilized after 122 h, when the fruit was still marketable, whereas in H2, the weight loss occurred from the very beginning when the fruit is still marketable. Similarly, the insulated box was highly effective at slowing down weight loss compared to the non-insulated box, but was more effective for fruits from the short-distance production area than for fruits from the long-distance production area. Time of harvest and lower temperature during transit is critical to reducing the weight loss of fruits.

3.3.2. Firmness

The average firmness of 'Nam Dok Mai Si Thong' mango fruit with peel decreased during storage, regardless of the distance of the mango production area from the tarmac, and the temperature difference experienced at the tarmac, as shown in Figure 4a. However, the average firmness decreased more rapidly in fruits from the long-distance production area (H2) than in those from the short-distance area (H1). Constant temperature (20 °C) slowed down the decrease in firmness of the fruit compared with insulated and non-insulated boxes exposed to the sun, but was more effective in fruit from the short-distance production area than from that from the long-distance production area. Insulated boxes were highly effective at decreasing firmness losses compared to non-insulated boxes, but were more effective in fruit from the short-distance production area than in fruit from the long-distance production area.

The peel toughness of 'Nam Dok Mai Si Thong' mango fruit increased during storage, regardless of the distance of the mango production area from the tarmac and the temperature difference experienced at the tarmac, as shown in Figure 4b, However, the peel toughness increased more rapidly in fruit from the long-distance production area (H2) than in fruit from the short-distance production area (H1). Constant temperature (20 °C) slowed down the increase in peel toughness of the fruit compared with insulated and non-insulated boxes exposed to the sun, but was more effective in fruit from the short-distance production area (H1) than in fruit from the long-distance production area (H2).

Destructive assessment of the fruit clearly showed the same trend as in the non-destructive assessment in the case of the peel's initial firmness, rupture force and average firmness (Figure 4c–e), but contradicted the non-destructive assessment in the case of peel toughness (Figure 4f). The average firmness of 'Nam Dok Mai Si Thong' mango fruit with peel decreased during storage regardless of the distance of mango production area from the tarmac and the temperature difference experienced at the tarmac. In addition, other parameters such as initial firmness, rupture force, and average firmness show a similar trend. However, the decreases in these parameters were much higher in fruit from the

long-distance production area (H2) than in fruit from the short-distance production area (H1). Constant temperature (20 °C) delayed the softening in both production areas as compared to the insulated and non-insulated boxes, but the delay was highly effective in fruit from the short-distance production area (H1).

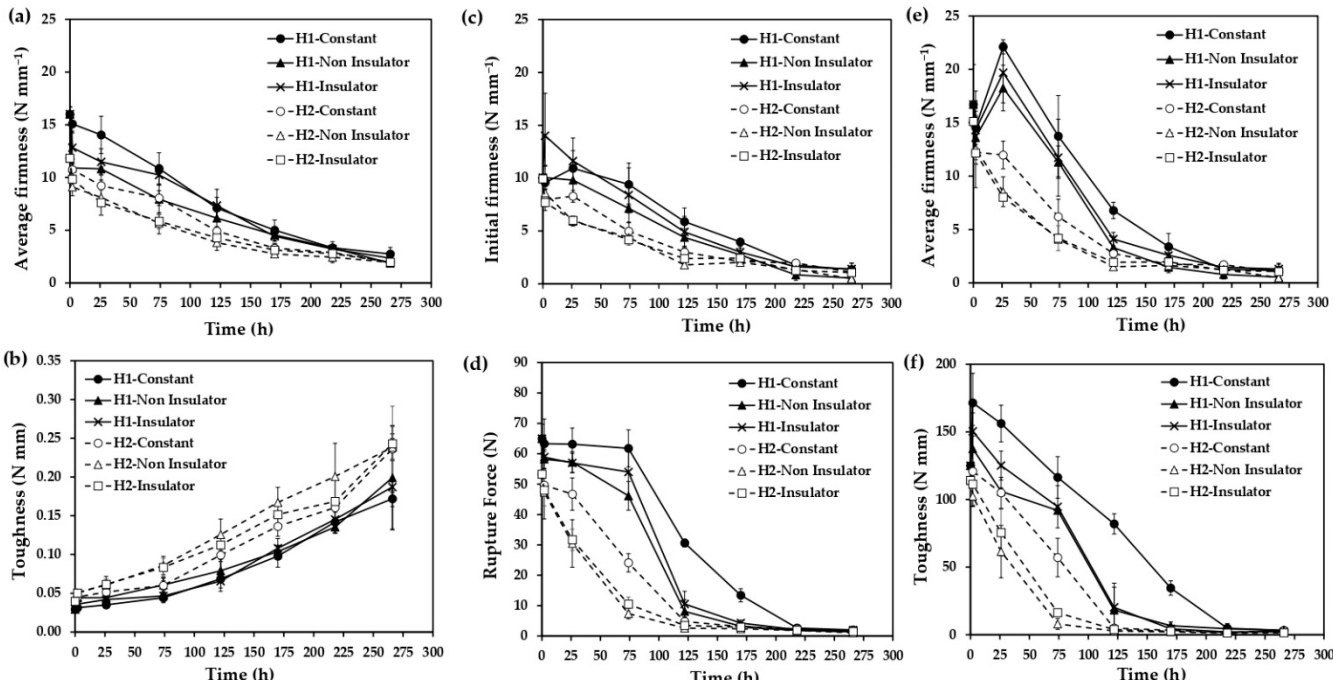

**Figure 4.** (**a**) Non-destructive—average firmness; (**b**) non-destructive—toughness; (**c**) destructive—initial firmness; (**d**) destructive—rupture force; (**e**) destructive—average firmness; and (**f**) destructive—toughness of mango fruit ('Nam Dok Mai Si Thong') under constant, non-insulated, and insulated conditions during simulated handling at the tarmac. After simulated handling at the tarmac, mango fruits from the first and second harvest were transferred to a temperature of 20 °C for 266 h. H1, first harvest; H2, second harvest. Vertical bars on the graph indicate S.D. (*n* = 3).

### 3.3.3. Total Soluble Solids (TSS), Acidity and pH

Total soluble solids (TSS) are important for enhancing the sweetness of the mango fruit. During ripening, TSS increases, acidity decreases, and pH increases. The total soluble solids of 'Nam Dok Mai Si Thong' mango fruit increased during storage regardless of the distance of the mango production area from the tarmac and the temperature difference experienced at the tarmac, as shown in Figure 5a. However, the TSS increased more rapidly in fruit from the long-distance production area (H2) than in fruit from the short-distance area (H1). Constant temperature (20 °C) slowed down the increase in the TSS of the fruit compared with insulated and non-insulated boxes exposed to the sun, but was more effective in fruit from the short-distance production area (H1) than in fruit from the long-distance production area (H2).

In contrast, acidity decreased gradually as the fruit matured, and a similar trend can be observed in Figure 5b regardless of the distance of production area from the tarmac or the temperature difference experienced at the tarmac. Constant temperature (20 °C) or insulation don't influence the reduction in acidity.

pH, conversely, gradually increased during storage from about 3 to 5 regardless of the distance of the production area from the tarmac and the temperature experienced at the tarmac, as shown in Figure 5c. However, the pH increased more rapidly in fruit from the long-distance area (H2) than in fruit from the short-distance area (H1). Constant temperature (20 °C) or insulation didn't influence the reduction in acidity.

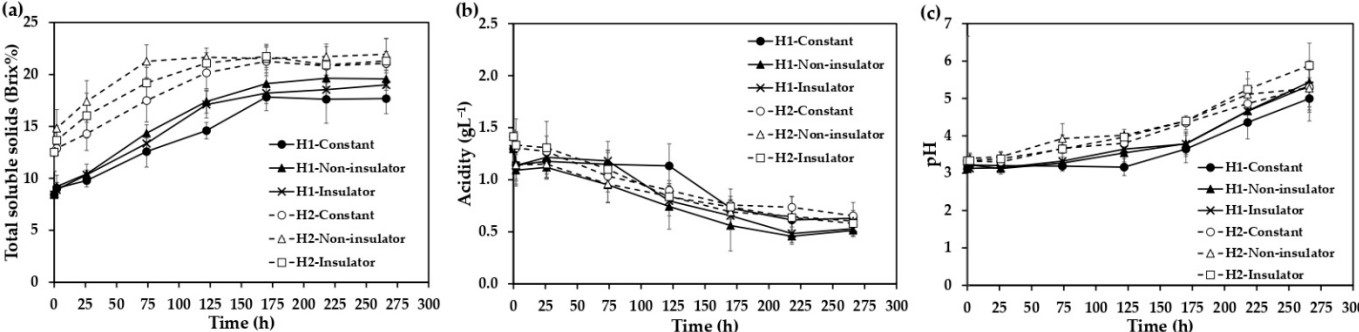

**Figure 5.** (**a**) Total soluble solids, (**b**) acidity, and (**c**) pH of mango fruits ('Nam Dok Mai Si Thong') under constant, non-insulated, and insulated conditions during simulated handling at the tarmac. After simulated handling at the tarmac, mango fruits from the first and second harvests were transferred to 20 °C for 266 h. H1, first harvest; H2, second harvest. Vertical bars on the graph indicate S.D. (*n* = 5).

### 3.3.4. Colour

The colour of the fruit is very important for the marketability of mango fruit, and is the first factor that attracts the consumer. This is more critical in cv. Nam Dok Mai Si Thong, because it has a yellow peel at both unripe and ripe stages. The *L\** value of peel of 'Nam Dok Mai Si Thong' mango fruit decreased during storage, regardless of the distance of the mango production area from the tarmac and the temperature difference experienced at the tarmac, as shown in Figure 6a. However, the *L\** value decreased more rapidly in fruit from the short-distance production area (H1) than in fruit from the long-distance area (H2). A higher decrease in *L\** value was found in fruit from the short-distance production area (H1). Constant temperature (20 °C) slowed down the decrease in *L\** value of the fruit compared with insulated and non-insulated boxes exposed to the sun. Conversely, the *L\** value of the fruit rapidly decreased in non-insulated boxes irrespective of the distance of the production area from the tarmac (Figure 7).

The *a\** and *b\** values of the peel of 'Nam Dok Mai Si Thong' mango fruit increased during storage regardless of the distance of the mango production area from the tarmac and the temperature difference experienced at the tarmac, as shown in Figure 6b,c. However, the values of *a\** and *b\** increased more rapidly in fruit from the long-distance production area (H2) than in fruit from the short-distance area (H1). The greater increase in the values of *a\** and *b\** observed in fruit from the long-distance production area (H2) could be due to longer time after harvest (70 h) resulting in higher temperature built up in the simulated boxes and a higher respiration rate. Constant temperature (20 °C) slowed down the increase in *a\** and *b\** values of the fruit compared with insulated and non-insulated boxes exposed to the sun, but was more effective in fruit from the short-distance production area (H1) than in fruit from the long-distance production area. Mango fruit peel in non-insulated boxes exposed to the sun turned red and yellow before fruits under constant and insulated conditions (Figure 7).

The *L\** value of the flesh near the seed of 'Nam Dok Mai Si Thong' mango fruit decreased during storage regardless of the distance of the mango production area from the tarmac and the temperature difference experienced at the tarmac, as shown in Figure 6e, However, the *L\** value of flesh decreased more rapidly in fruit from the long-distance production area than in fruit from the short-distance area. Constant temperature (20 °C) slowed down the decrease in the *L\** value of the fruit compared with insulated and non-insulated boxes exposed to the sun (Figure 8).

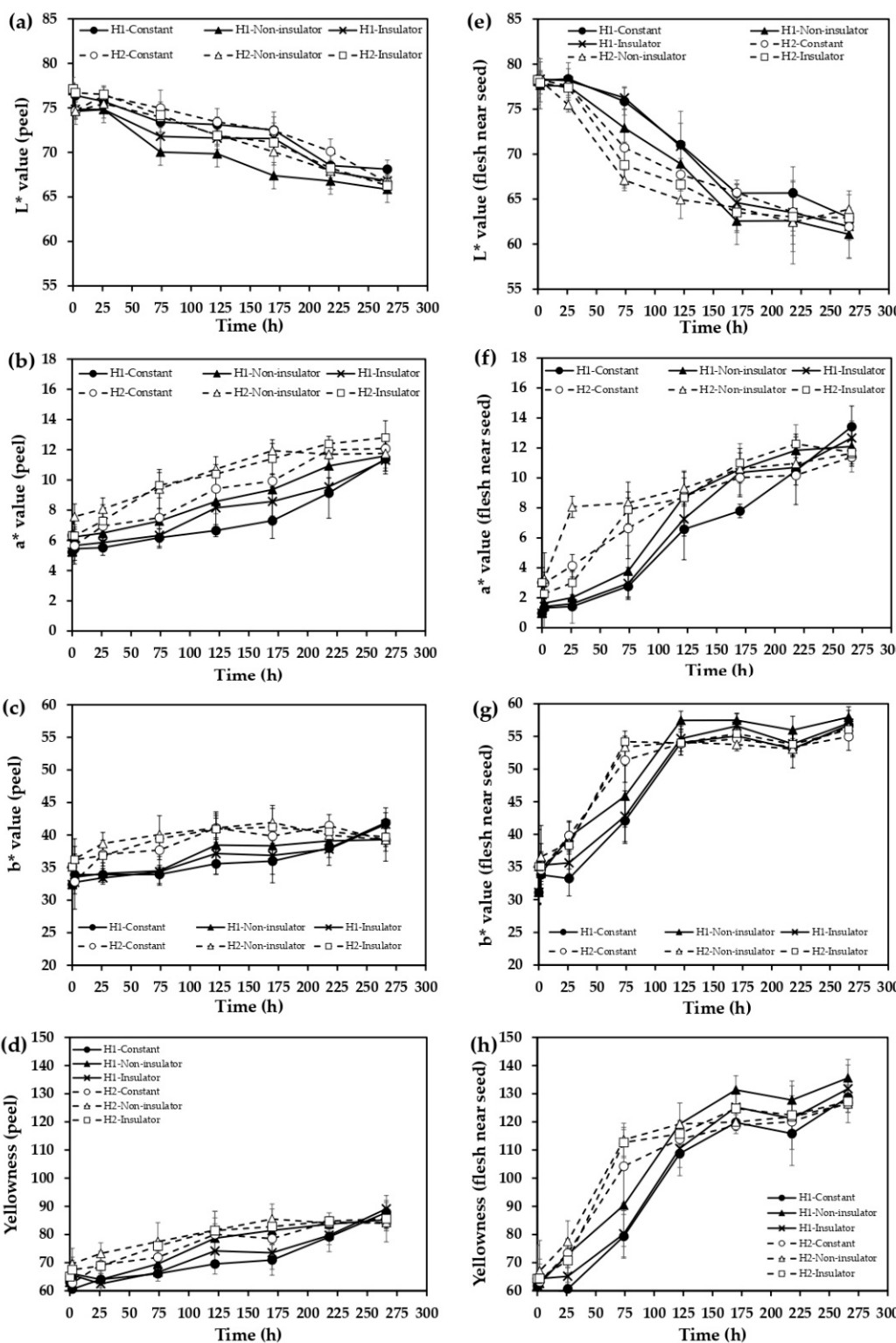

**Figure 6.** (**a**) *L\**, (**b**) *a\**, (**c**) *b\** value, and (**d**) yellowness of the peel of mango fruits; (**e**) *L\**, (**f**) *a\**, (**g**) *b\** value, and (**h**) yellowness of the flesh near the seed of mango fruits ('Nam Dok Mai Si thong') under constant, non-insulated, and insulated conditions during simulated handling at the tarmac. After simulated handling at the tarmac, mango fruits from the first and second harvest were transferred to 20 °C for 266 h. H1, first harvest; H2, second harvest. Vertical bars on the graph indicate S.D. (*n* = 3).

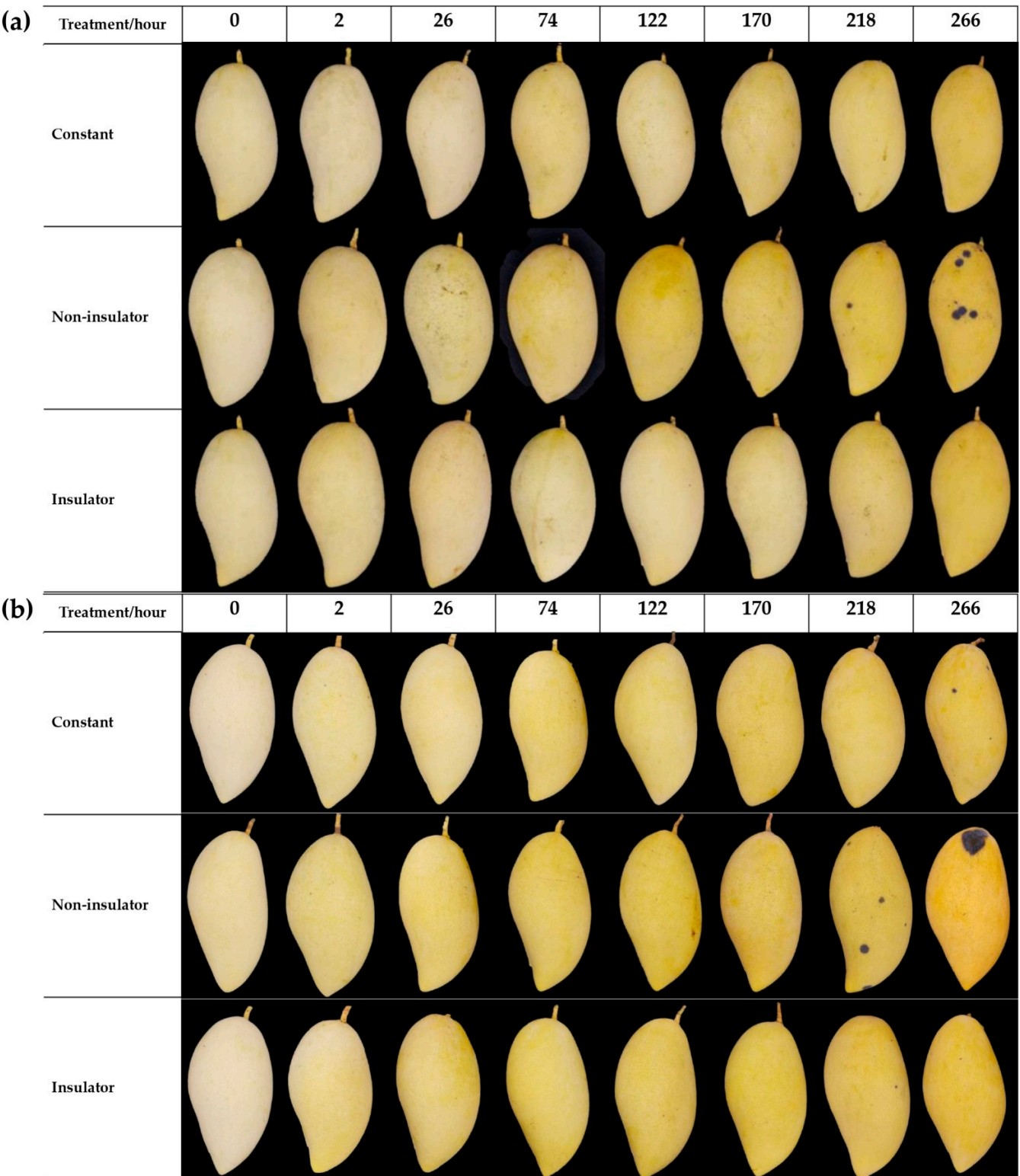

**Figure 7.** Colour of the peel of mango fruits ('Nam Dok Mai Si Thong') from the (**a**) first harvest, and (**b**) second harvest under constant, non insulated, and insulated conditions during simulated handling at the tarmac. After simulated handling at the tarmac, mango fruits were transferred to 20 °C for 266 h.

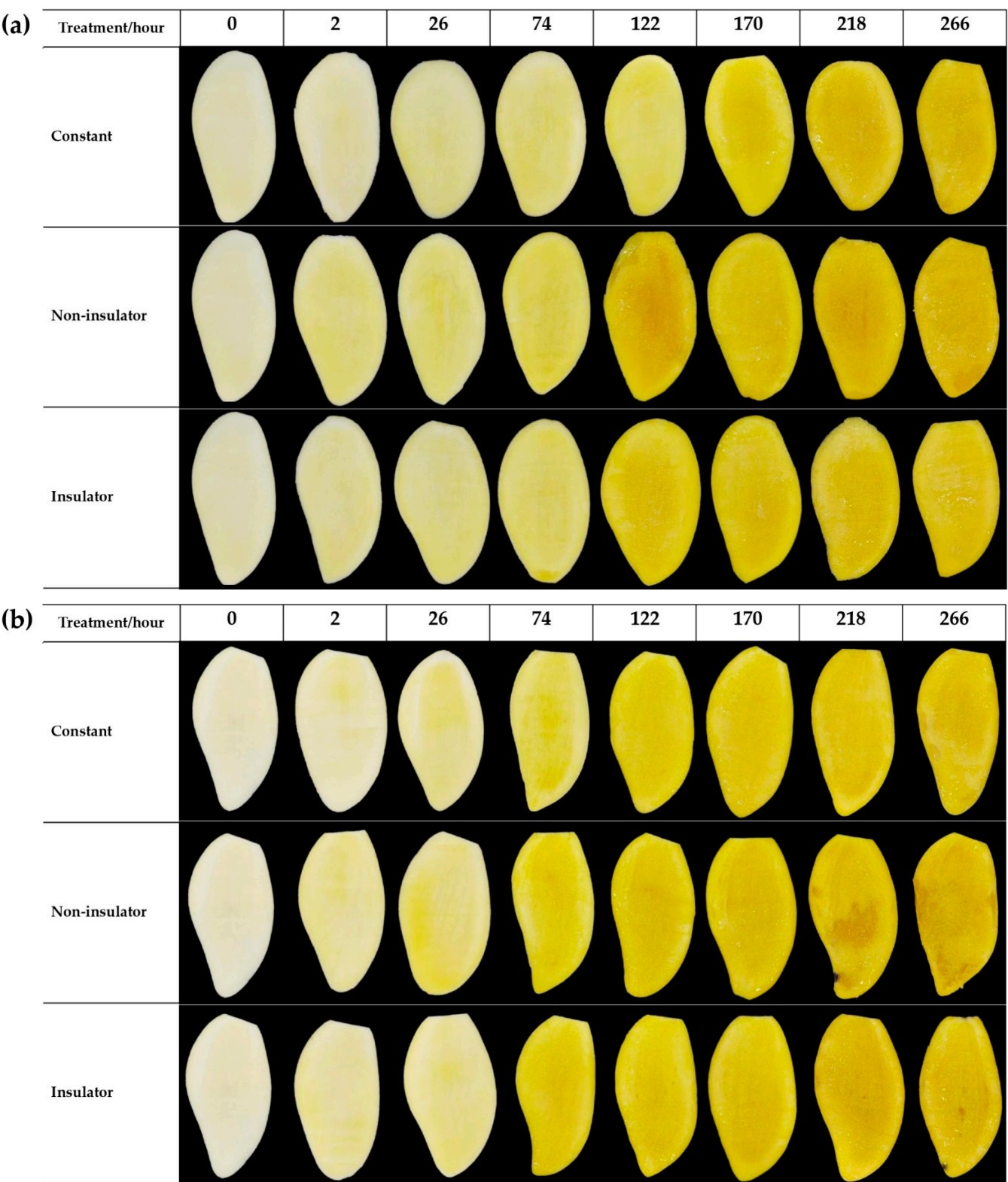

**Figure 8.** Colour of the flesh near the seed of mango fruits ('Nam Dok Mai Si Thong') from the (**a**) first harvest and (**b**) second harvest under constant, non-insulated, and insulated conditions during simulated handling at the tarmac. After simulated handling at the tarmac, mango fruits were transferred to 20 °C for 266 h.

The *a** and *b** values of the flesh near the seed of 'Nam Dok Mai Si Thong' mango fruit increased during storage regardless of the distance of the mango production area from the tarmac and the temperature difference experienced at the tarmac, as shown in Figure 6f,g, However, the values of *a**, and *b** increased more rapidly in fruit from the long-distance production area than in fruit from the short-distance area. Constant temperature (20 °C) slowed down the increase in the *a** and *b** value of the fruit compared with insulated and non-insulated boxes exposed to the sun, but was more effective in fruit from the short-distance production area than in fruit from the long-distance production area. Mango fruit in non-insulated boxes exposed to the sun turned red and yellow on flesh near the seed before fruit under constant or insulated conditions.

The yellowness of peel and the flesh near the seed of 'Nam Dok Mai Si Thong' mango fruit increased during storage regardless of the distance of the mango production area from the tarmac and the temperature difference experienced at the tarmac, as shown in Figure 6d,h. However, the yellowness value increased more rapidly in fruit from the long-distance production area (H2) than in fruit from the short-distance area (H1). The greater increase in the yellowness of fruit from the long-distance production area could be due to the longer time after harvest (70 h) resulting in higher temperature built up in the simulated boxes. Constant temperature (20 °C) slowed down the increase in the yellowness of the fruit compared with insulated and non-insulated boxes exposed to the sun, but was more effective in fruit from the short-distance production area (H1) than in fruit from the long-distance production area (H2).

### 3.3.5. Visual Assessment

The peel colour of mangoes changes from light yellow in mature fruit to dark yellow or golden yellow in ripe mangoes. The visual score of yellowness increased steadily during storage, irrespective of the distance of the production area from the tarmac or the difference in temperature experienced at the tarmac, as shown in Figure 9a. Yellowness increased rapidly in fruit from the long-distance production area (H2) compared to in fruit from the short-distance production area (H1). Similarly, yellowness increased most rapidly under non-insulated conditions, followed by insulated and constant.

Shrivelling occurred earlier (75 h under constant conditions) in fruit from the long-distance mango production area (H2) compared to in fruit from the short-distance mango production area (H1) (125 h under constant conditions), as shown in Figure 9b. However, the signs of shrivelling developed in the following order: non-insulated, insulated, and constant conditions.

Decay tended to increase during storage, but it never attained the critical value of rejection (Figure 9c). However, the decay score was higher in fruit from the long-distance production area (H2) than in fruit from the short-distance production area (H1). Similarly, the score was lower under constant or insulated conditions than in non-insulated conditions.

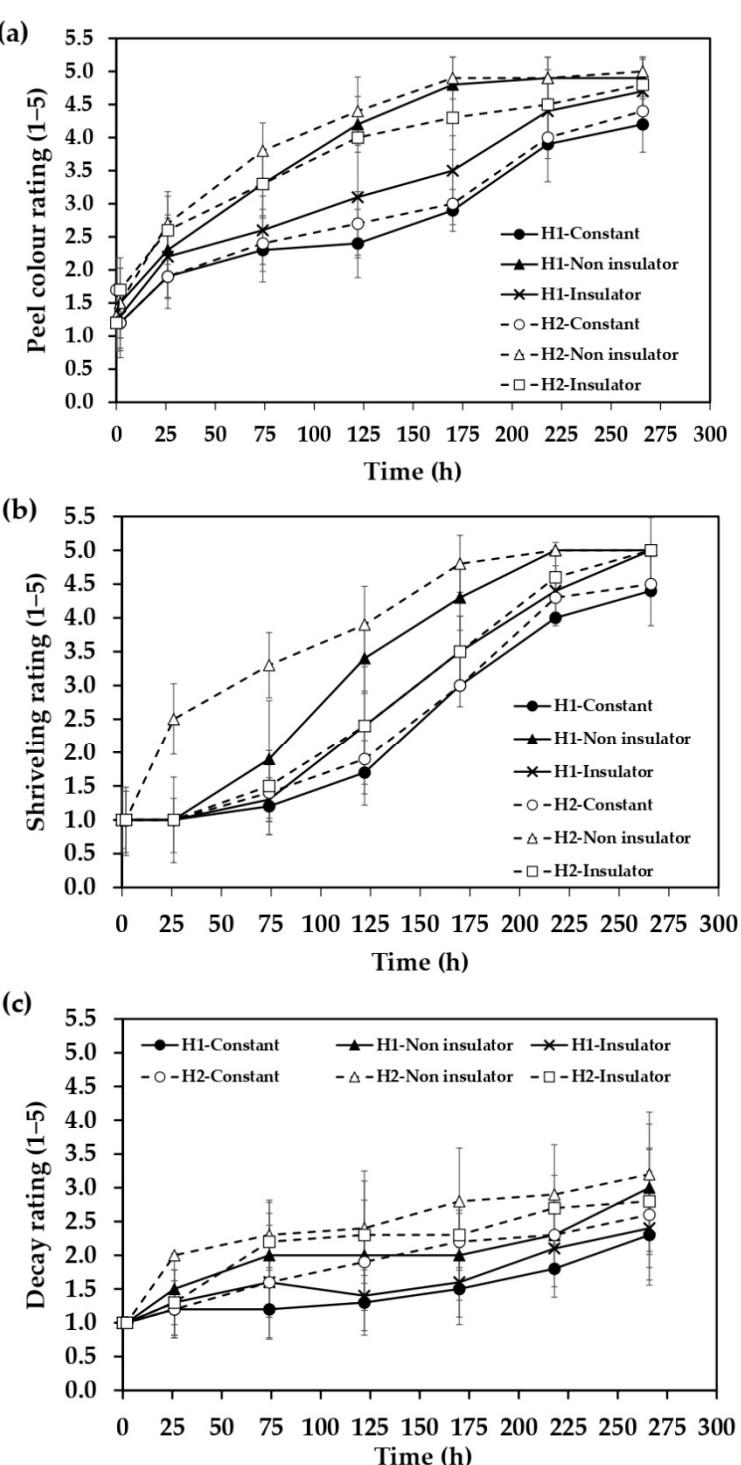

**Figure 9.** (**a**) Peel colour, (**b**) shrivelling, and (**c**) decay of mango fruits ('Nam Dok Mai Si thong') under constant, non-insulated, and insulated conditions during simulated handling at the tarmac. After simulated handling at the tarmac, mango fruits from the first and second harvest were transferred to 20 °C for 266 h. H1, first harvest; H2, second harvest. Vertical bars on the graph indicate S.D. (*n* = 10).

## 4. Discussion

### 4.1. Temperature and Relative Humidity Monitoring

Temperature is a critical factor for the postharvest shelf life and quality of mango fruit, and therefore, an effective cold chain is an integral part of the postharvest handling of perishable fruits. This is more important when perishable produce is shipped to far distant

markets. However, the cold chain is interrupted for a few to several hours at the tarmac of the airport, causing irreparable damage to the produce. Insulating the mango boxes with polystyrene fibre was shown to be effective at reducing the temperature built up by 8.8 °C (H1) and by 3.8 °C (H2) when exposed to the sun for 2 h. The reduction in temperature was more prominent in from the short-distance production area (H1) than in fruits from the long-distance production area (H2). The length of time following harvest in fruits from the long-distance production area (H2) could have triggered higher respiration, resulting in higher temperature being built up in the boxes.

These results are consistent with a report by Pelletier et al. [16], who stated that this results in an increase in the average internal temperature of the product to 13 °C. Our data are also consistent with the report of Bollen and others [41], who measured the temperature inside pallets of asparagus sent by air from Auckland to Tokyo. The temperature of the asparagus jumped from 4 °C to 14 °C within 30 min of ground operations in Auckland. Additionally, during the loading and unloading of the aircraft from South Africa to Spain, Abad et al. [42] reported that the temperature in fresh fish increased by 5 °C. In addition, Pelletier et al. [43] reported an increase in temperature from 6 °C to approximately 9–16 °C on each side of the ULD wall when the instrumented container was on the tarmac for 1 h 50 min.

### 4.2. Shelf Life

The long shelf life of mango fruit is one of the most important characteristics that are desirable for commercial purposes. Insulation of mango with polystyrene fibre was effective for extending the marketable shelf life of mango fruits by 48 h. The effectiveness of LDPE and constant temperature (20 °C) and LDPE and insulation is unaffected by short (H1) or long distance (H2), with shelf life increasing by 48 h in either case. This clearly proves that use of LDPE and insulation in ULD negates the negative impact of abusive temperature conditions on the tarmac during air freight handling. Insulation lowers the temperature in the box, which inhibits respiration and slows weight loss, resulting in longer shelf life for the fruit. However, the benefit is much higher when the mango production area is close to the airport. The results of the present study parallel those reported by Nunes et al. [13], who reported that papayas handled under semi-constant temperature regimes had a longer shelf life than papayas handled in regimes with fluctuating temperatures. Similarly, strawberries that had spent a 10 h in a fluctuating-temperature regime were deemed unmarketable prior to retail, whereas those stored at a semi-constant temperature (10 h continuously at 3 °C) were still acceptable after 24 h in the retail display [44]. This is also consistent with the results of studies on Snap beans, where those subjected to fluctuating temperature were considered unmarketable prior to exposure to retail conditions, while those subjected to a constant temperature treatment were no longer acceptable only after spending 24 h in the retail display [45].

### 4.3. Quality Assessment
### 4.3.1. Weight Loss

The primary predictor of mango fruit quality and storage time is weight loss, which is correlated to the respiration rate and transpiration of the fruit [46]. Temperature is a critical factor influencing weight loss in fruits. Constant temperature (20 °C) effectively slowed down weight loss because of the lower temperature, probably by inhibiting respiration, although H1 and H2 responded differently. A higher effectiveness was found in H1 than in H2, in which the fruits were from the long-distance production area. Similarly, insulated boxes were highly effective at slowing down weight loss compared to non-insulated boxes, but were more effective in fruit from the long-distance production area than in fruit from the short-distance production area. The insulated boxes may have been more effective for H1 (28 °C) than H2 (34.2 °C) because of the lower temperature in H1 and the more physiologically mature fruits (72 h after harvest) in H2 compared to in H1 (53 h). The

amount of time after harvesting and lower temperatures during transit are critical to reducing weight loss in fruits.

The results obtained for weight loss in mango are in agreement with the values previously reported by Nunes and Emond. They reported that weight loss of 'Tommy Atkins' and 'Palmer' mangoes stored at 20 °C reached about 1.5% of the fruit's initial weight after 2 days, and weight loss reached about 4.0% of the fruit's initial weight after 5 days [47]. In addition, Noiwan et al. reported weight loss of around 12% in 'Nam Dok Mai Si Thong' mango stored at 20 °C for 12 days and stored at 34 °C for 5 days [8]. Weight loss in fresh fruits and vegetables is mostly caused by water loss from transpiration and respiration processes [48]. Water loss is another important element affecting mango fruit quality [49].

### 4.3.2. Firmness

The firmness of fruit is an essential quality parameter, and delaying the decrease in firmness increases the shelf life of fruit. The greater decrease in firmness found in fruit from the long-distance production area (H2) could be due to longer time after harvest (70 h after harvest and simulated airport handling treatment) resulting in a higher temperature being built up in the simulated boxes, and probably a higher respiration rate.

Insulated boxes were highly effective at decreasing firmness compared to non-insulated boxes, but this was more pronounced in fruit from the short-distance production area than in fruit from the long-distance production area. This is consistent with reports from previous studies, with Nunes et al. [13] observing that the firmness of papaya stored at a semi-constant temperature was more acceptable than papaya subjected to fluctuating cold and fluctuating warm temperature regimes during simulated handling. Jha et al. [50] reported that the peel firmness of mango cultivars varied from the initial level of 13.4–27.1 N to 3.9–24.5 N during the ripening period of 10 days. Yasunaga et al. [7] reported that the mango 'Nam Dok Mai' softened during storage following the actual distribution temperature profile from Thailand to Japan.

The higher increase in toughness observed in fruit from the long-distance production area could be due to the longer time after harvest (70 h after harvest and simulated airport handling treatment) resulting a higher temperature being built up in the simulated boxes, as well as a higher respiration rate. Constant temperature (20 °C) slowed down the increase in peel toughness of the fruit compared with insulated and non-insulated boxes exposed to the sun, but this was more pronounced in fruit from the short-distance production area than in fruit from the long-distance production area. Peel toughness increases during storage, as reported by Jha et al. [50]. In general, the peel toughness of early-harvested mangoes was found to be higher throughout the ripening period in all cultivars. This increase in the first days of ripening could be explained by differences in maturity and ripeness [51].

The destructive assessment of the fruit clearly showed the same trend as revealed by the non-destructive assessment in the case of peel firmness, rupture force, and initial and average firmness, but contradicted those results in the case of peel toughness. Decreased firmness in mango fruits has been reported by Noiwan et al. [8], who described a loss of firmness in mango from about 48 N on the initial day of storage to approximately 10 N within 4 days of storage at 20 °C. Similar findings have been reported by Jha et al. [50], Ketsa et al. [52], Nadeem et al. [53]. The decrease in firmness is caused by the degradation of cell wall constituents and polysaccharides caused by the action of polygalacturonase and pectin esterase on the solubilization of pectin substrates [54–56]. Fruit firmness is an essential factor for the postharvest fruit quality [11]. The shelf life and supply chains of mango fruit are significantly shortened by accelerated ripening and softness. The genetically intricate process of mango fruit ripening is brought on by an increase in ethylene production and respiration rate [57].

### 4.3.3. Total Soluble Solids (TSS), Acidity, and pH

Mango fruit ripening is associated with an increase in TSS, a decrease in acidity, and an increase in pH. Mango cv. Nam Dok Mai Si Thong also follows the same trend, and these behaviours were observed in fruit from both the short-distance production area and the long-distance production area. However, early increases were observed in fruit from the long-distance production area and fruits that were not insulated. The higher increase in the TSS of fruit from the long-distance production area could be due to the longer time after harvest (70 h) resulting in higher temperature being built up in the simulated boxes and a higher respiration rate. Increased TSS in mango fruits has also been reported by Noiwan et al. [8]. In addition, Yasunaga et al. [7] reported that the TSS of the 'Nam Dok Mai' mango from Phitsanulok fruit after distribution increased by about 1.8 times compared to the fruits after harvest, but that it gradually decreased with time, in a manner dependent on the storage temperature. A mango's TSS initially increases, then decreases, and finally reaches a stage of full senescence [58].

The acidity of 'Nam Dok Mai Si Thong' mango fruit decreases gradually during storage and is not influenced by the distance of the production area from the tarmac, temperature, or insulation. Decreased acidity during fruit ripening is connected to the transformation of organic acids into sugars [59]. Due to the conversion of citric acid into sugars and their subsequent usage in the fruit's metabolism, particularly during the respiration process, the concentration of organic acids in ripe mango fruits decreases throughout ripening [8,60].

The pH of 'Nam Dok Mai Si Thong' mango fruit increased during storage as the fruit ripened, and was influenced by the distance from the production area (70 h), but not by temperature or insulation. These results are consistent with those presented in a report by Islam et al. [61].

### 4.3.4. Colour

The colour of Nam Dok Mai Si Thong mango fruit is the most critical characteristic in consumers' eyes, and the development of colour increases with ripening. The decrease in $L^*$ value was greater in fruit from the short-distance production area (H1), whereas the values of $a^*$ and $b^*$ increased in fruit from both the short- and long-distance production areas (H2). The yellowness of Nam Dok Mai Si Thong is influenced by the increase in $b^*$, which was observed in more mature fruits. During mango fruit ripening, mango fruits gather phenolic and carotenoid chemicals [62]. Carotenoids are the main pigments responsible for the yellow-to-orange colour of mature mango fruit [63]. The values of $L^*$, $a^*$, and $b^*$ were consistent with those reported by Noiwan et al. [8], Nunes et al. [40], Nadeem et al. [53]. Changes in the colour of the mango peel occur because of both chlorophyll degradation and carotenoid synthesis during ripening [64].

One of the most accurate measures of maturity or ripeness is the colour of the flesh. The $L^*$ value decreased and the $a^*$ and $b^*$ values of the flesh increased irrespective of the distance of the production area from the tarmac and the temperature. Unlike in peel, flesh colour in fruit from the long-distance production area showed a decrease in $L^*$ value but an increase in $a^*$ and $b^*$ values. The yellow flesh colour starts to appear from the seed out as the fruit matures more. Following harvest, the mango ripens, and more of the interior is covered in a deeper shade of colour. The spectrum of colours varies by cultivar [65].

### 4.3.5. Visual Assessment, Shrivelling and Decay

The colour of mango peel has a significant impact on how appealing it is to consumers, because fruit with more red blush and no green tint is frequently more expensive or easier to sell [66]. Yellowness increased earlier in fruit from the long-distance production area (H2) than in fruit from the short-distance production area (H1), which was similar to the colour estimation, suggesting that both techniques provide similar outcomes. Therefore, in the present study, the yellow colour observed in the mango fruit exposed without insulation to 36.9 ± 3.2 °C (H1) and 38.0 ± 1.3 °C (H2) for 2 h during simulated airport operations and flight may have been a result of the exposure to high temperature during the handling of the

mango. Shrivelling occurred earlier in fruit from the long-distance production area (H2) and in non-insulated boxes exposed to the sun, which could be due to the higher temperature built up during the simulation. Interestingly, decay at critical value for rejection was not attained by any of the treatments. However, decay occurs more in mangoes subjected to a fluctuating regime than those at constant temperature. Nunes and Emond previously reported that the storage of mangoes under fluctuating temperatures during handling could cause water condensation on commodity surfaces, potentially increasing the development of decay due to fungal and bacterial pathogens [22]. This is consistent with the findings presented in the study by Nune et al. [18], which found that papayas handled in fluctuating temperature regimes developed an objectionable colour, were more shrivelled, and had more decay than papayas handled under semi-constant temperature regimes.

## 5. Conclusions

The results of this study indicate that the temperature at the tarmac and the distance of mango production area from the tarmac significantly affect the quality of mango fruit. Exposure to high temperatures for a short period of 2 h resulted in a reduction in shelf life, accelerated weight loss, decreased fruit firmness, etc. In addition, it proved that high temperature, which is frequently encountered during air cargo handling, can cause loss of quality. This could result in a significant amount of mango fruit being rejected upon arrival at the destination, even if the exposure to the abusive temperature at the tarmac is of short duration (i.e., 2 h at more than 36 °C). However, this research also showed that the insulation of mango fruit boxes with polystyrene fibre mitigates the deleterious effect of exposure to 2 h direct sun by reducing the temperature increase, thus extending the shelf life and quality of mango fruit. Shelf life was increased by 48 h under both insulated and constant conditions in fruit from both the short- and long-distance production areas. Insulation was also effective at delaying fruit weight loss, delaying the decrease in fruit firmness, delaying the decrease in rupture force, and maintaining the yellow colour of the peel and pulp. This paper also provided support for the advantageous nature of locating mango production areas close to an airport.

**Author Contributions:** Conceptualization, P.P. and P.S.; methodology, K.S., P.P., P.S. and N.C.; software, T.P. and S.R.; validation, U.K.P., P.P. and P.S.; formal analysis, K.S. and P.M.; investigation, K.S.; resources, P.P. and P.S.; data curation, K.S., C.C. and P.M.; writing—original draft preparation, K.S.; writing—review and editing, U.K.P. and W.K.; visualization, K.S.; supervision, P.P.; project administration, K.S.; funding acquisition, P.P. All authors have read and agreed to the published version of the manuscript.

**Funding:** This work was financially supported by King Mongkut's Institute of Technology Ladkrabang Research Fund.

**Data Availability Statement:** Not applicable.

**Acknowledgments:** We would like to acknowledge Program Management Unit for Competitiveness (PMUC), Office of National Higher Education Science Research and Innovation Policy Council, Thailand and, AOT TAFA Operator Company Limited. This research project was supported by King Mongkut's Institute of Technology Ladkrabang Research Fund.

**Conflicts of Interest:** The authors declare no conflict of interest. The funders had no role in the design of the study; in the collection, analyses, or interpretation of data; in the writing of the manuscript; or in the decision to publish the results.

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
