# Peer review of "Temperature Difference in Loading Area (Tarmac) during Handling of Air Freight Operations and Distance of Production Area Affects Quality of Fresh Mango Fruits (Mangifera indica L. ‘Nam Dok Mai Si Thong’)"

_horticulturae, doi:10.3390/horticulturae8111001_

Round 1
Reviewer 1 Report
The topic is interesting but it is badly organized.
Tittle:
1. It is too difficult to understand and must be reorganized.
Introduction:
1. There are seven paragraphs in this section, too many. The key information in each paragraph is quite missing, some of them are not necessary.
Materials:
1. Line 326: correct "Figure 2 " with "Figure 3".
2. there are one table and three figures in this section, it is quite strange. I suggest condensing them into one figure.
Results:
Major concerns:
There are a total of 13 figures, too many. I suggest condensing them into five or six figures.
Line 371: the table title is "Shelf life of ....", while no shelf life data were present in the table.
Line 411: Figure 4 is not the result of the present study.
Minor concerns:
Line 453: Table 4: "ns means non-significant", so what does it mean when the same letter is used? To use the different letter indicating significance, or use * indicating significance (plus ns indicating non-significant). But do not mix using.
Discussion:
This section is not well organized, and can not be called "Discussion". I suggest authors focus on the main conclusion of the present study and further highlight the significance of their contributions to the field.
Author Response
Comments and Suggestions for Authors
The topic is interesting but it is badly organized.
Tittle:
- It is too difficult to understand and must be reorganized.
-I reorganized and changed proposed it to make it more understandable.
Introduction:
- There are seven paragraphs in this section, too many. The key information in each paragraph is quite missing, some of them are not necessary.
-I reorganized as you suggested.
Materials:
- Line 326: correct "Figure 2 " with "Figure 3".
- I have edited successfully. Before editing the new manuscript, I removed figure 2 from the previous manuscript. (Line 282)
- there are one table and three figures in this section, it is quite strange. I suggest condensing them into one figure.
-I removed some figure before editing.
Results:
Major concerns:
There are a total of 13 figures, too many. I suggest condensing them into five or six figures.
-I reorganized as you suggested.
Line 371: the table title is "Shelf life of ....", while no shelf life data were present in the table.
-I’ve already changed the table title. (Line 335-336)
Line 411: Figure 4 is not the result of the present study.
-This is result of this study
Minor concerns:
Line 453: Table 4: "ns means non-significant", so what does it mean when the same letter is used? To use the different letter indicating significance, or use * indicating significance (plus ns indicating non-significant). But do not mix using.
- I already edited it as shown in Table 4(a,b).
Discussion:
This section is not well organized, and can not be called "Discussion". I suggest authors focus on the main conclusion of the present study and further highlight the significance of their contributions to the field.
- I already edited it as you suggested.
Extensive editing of English language and style required
- has undergone English language editing by MDPI
"Please see the attached manuscript, which has been edited in accordance with your suggestions."

Reviewer 2 Report
Srisawat investigate in their manuscript the impact of temperature exposure as occurring during transportation and shipping on the shelf life of Mango. They simulated these processed with two sets of samples and subsequently analysed quality parameters like firmness, TSS and acidity. The manuscript is logically structured and the experiments well executed. However, post-harvest physiology of mango has been studied in the past intensively. Thus, the results are not too surprising. However, the data might be interesting for those working in that field
In addition, there are some points that must be addressed:
*) Results: at the beginning of each sub-section the authors should say a few words about the experiment and/or its setup. In the current version the authors start directly with the obtained data, which makes it necessary to look to the results section. This makes the manuscript difficult to read. Thus, it would be good if the authors could summarize the aim of each experiment at the beginning of each paragraph.
*) The title of Table 2 is misleading since no data for shelf life are shown in that table.
*) The data in Table 2 data are difficult to interpret since only average temperatures are shown rather than temp. fluctuations (like in Fig. 4).
*) It is difficult to understand how the temperature in the "Insulator" experiment can fist increase to approx. 30 °C and then decline to approx. 28°C and then stay relatively constant while the outside temperature is much higher and exceeds 40°C. Do the authors have any explanation for that?
*) There is no figure 3.
*) Line 417: the authors should be more precise for the terms “harvest time” and “time needed for transportation”
*) Table 4: Statistical tests: in the caption the authors state that “…letters indicate the significantly different in each columns within each harvest …”. Thus, I think the authors should also use a, b … in the lower panel rather than e, d, … If the authors think that this is too confusing they may make separate.
*) Figure 5: the unit should be given N mm-1 rather than N/mm (this applies for the whole manuscript.
*) Figure 7: what is the meaning of % acidity? Acidity is usually given in mmol/l; if it is given in % the acid used for its calculation must be stated. In addition, I do not think that the method used (electrical conductivity) is a reliable criterion for measurement of acidity. It would be much better to titrate aliquots of the samples.
*) Figures 8 and 9 should be combine and the axes should be scaled in the same way.
*) The conclusion should be more detailed.
*) The English needs to be improved throughout the manuscript.
Author Response
Comments and Suggestions for Authors
Srisawat investigate in their manuscript the impact of temperature exposure as occurring during transportation and shipping on the shelf life of Mango. They simulated these processed with two sets of samples and subsequently analysed quality parameters like firmness, TSS and acidity. The manuscript is logically structured and the experiments well executed. However, post-harvest physiology of mango has been studied in the past intensively. Thus, the results are not too surprising. However, the data might be interesting for those working in that field
In addition, there are some points that must be addressed:
*) Results: at the beginning of each sub-section the authors should say a few words about the experiment and/or its setup. In the current version the authors start directly with the obtained data, which makes it necessary to look to the results section. This makes the manuscript difficult to read. Thus, it would be good if the authors could summarize the aim of each experiment at the beginning of each paragraph.
- I already edited it as you suggested.
*) The title of Table 2 is misleading since no data for shelf life are shown in that table.
-I’ve already changed the table title. (Line 335-336)
*) The data in Table 2 data are difficult to interpret since only average temperatures are shown rather than temp. fluctuations (like in Fig. 4).
-we may need the table 2 temperature and humidity data.
*) It is difficult to understand how the temperature in the "Insulator" experiment can fist increase to approx. 30 °C and then decline to approx. 28°C and then stay relatively constant while the outside temperature is much higher and exceeds 40°C. Do the authors have any explanation for that?
-I removed this sentence.
*) There is no figure 3.
- I have edited successfully.
*) Line 417: the authors should be more precise for the terms “harvest time” and “time needed for transportation”
- I have edited successfully.
*) Table 4: Statistical tests: in the caption the authors state that “…letters indicate the significantly different in each columns within each harvest …”. Thus, I think the authors should also use a, b … in the lower panel rather than e, d, … If the authors think that this is too confusing they may make separate.
- I already edited it as shown in Table 4.(Line 382-390)
*) Figure 5: the unit should be given N mm-1 rather than N/mm (this applies for the whole manuscript.
- I already edited it as shown in Figure4. In addition, I applies for the whole manuscript.
*) Figure 7: what is the meaning of % acidity? Acidity is usually given in mmol/l; if it is given in % the acid used for its calculation must be stated. In addition, I do not think that the method used (electrical conductivity) is a reliable criterion for measurement of acidity. It would be much better to titrate aliquots of the samples.
- I already edited it as shown in Figure5.
-I chose this measuring tool because I needed to take a quick measurement.
-I also refer to the method described in the manuscript.
*) Figures 8 and 9 should be combine and the axes should be scaled in the same way.
- I already edited it as shown in Figure6.
*) The conclusion should be more detailed.
- I already edited it as you suggested.
*) The English needs to be improved throughout the manuscript.
- has undergone English language editing by MDPI
"Please see the attached manuscript, which has been edited in accordance with your suggestions."

Reviewer 3 Report
Dear Editor, in the manuscript horticulturae-1954502 authors evaluated the effect of distance of production area and different temperature conditions at the loading area (tarmac) on the quality of fresh mango fruits. Treatments were: Short distance (53h after harvest) and long distance 24 (70h after harvest) and three temperature conditions for 2h (simulated handling in tarmac) such as constant temperature (20 ºC), non-insulated or insulated and exposed to sun. In general, the experiment was well performed and the manuscript is well written providing interesting information showing that Insulation of mango fruit boxes mitigates the deleterious effect of exposure to 2h direct sun by reduction in the increase of the temperature and improves shelf life and quality of mango fruit.
Thus, the manuscript could be suitable for publication, although the following comments should be considered in the revision version:
- Lines 148-149: Please, clarify if the fruit harvested in these two dates were in the same ripening stage.
- Line 80: 10 fruits per treatment could be no enough given the high variability among individual fruits. Was the experiment replicate? This is an important issue that should be clarified before acceptance of the manuscript.
- Lines 234, 273: 3 fruits are a very low number of fruits. Please, clarify7.
- Line 335: Section 3 should be “Results”
- Line 371: It should be temperature and RH instead pf “shelf life”
- Data sho3wn in Table 2 seem to be also shown in figure 4, so that table 2 could be no necessary.
- Line 433: Weight loss is not shown in Figure 5.
- Line 435: No “much higher”
- Lines 447-449: These differences cannot be observed in Table 4. The reduction seems to be similar for 1st and 2nd harvests.
- Line 468: Consider adding “losses” after firmness.
- Figures: Significant differences among treatments or LSD values should be added to figures.
- Lines 1023, 1047, 1062: Provide reference number.
- Line 1026: Check H2 is repeated twice.
- Lines 1086 and 1097: Check these sentences to clarify.
- Use italic font for scientific names in reference list.
Author Response
Comments and Suggestions for Authors
Dear Editor, in the manuscript horticulturae-1954502 authors evaluated the effect of distance of production area and different temperature conditions at the loading area (tarmac) on the quality of fresh mango fruits. Treatments were: Short distance (53h after harvest) and long distance 24 (70h after harvest) and three temperature conditions for 2h (simulated handling in tarmac) such as constant temperature (20 ºC), non-insulated or insulated and exposed to sun. In general, the experiment was well performed and the manuscript is well written providing interesting information showing that Insulation of mango fruit boxes mitigates the deleterious effect of exposure to 2h direct sun by reduction in the increase of the temperature and improves shelf life and quality of mango fruit.
Thus, the manuscript could be suitable for publication, although the following comments should be considered in the revision version:
- Lines 148-149: Please, clarify if the fruit harvested in these two dates were in the same ripening stage.
- I added the phrase “and at the commercial harvesting stage” in Line 111
- Line 80: 10 fruits per treatment could be no enough given the high variability among individual fruits. Was the experiment replicate? This is an important issue that should be clarified before acceptance of the manuscript.
- Lines 234, 273: 3 fruits are a very low number of fruits. Please, clarify7.
-I think we didn’t replicate this experiment. However, the tarmac temperature conditions were almost identical and harvest maturity was same. The temperature effect was observed.
- Line 335: Section 3 should be “Results”
- I have edited successfully. (Line 292)
- Line 371: It should be temperature and RH instead pf “shelf life”
-I’ve already changed the table title. (Line 335-336)
- Data sho3wn in Table 2 seem to be also shown in figure 4, so that table 2 could be no necessary.
-we may need the table 2 temperature and humidity data.
- Line 433: Weight loss is not shown in Figure 5.
- I have edited successfully. The phrase "and Figure 5" was removed.” (Line 362)
- Line 435: No “much higher”
- I already edited it. (Line 362-363)
- Lines 447-449: These differences cannot be observed in Table 4. The reduction seems to be similar for 1st and 2nd harvests.
- I have edited successfully
- Line 468: Consider adding “losses” after firmness.
- I already edited it as shown in Line 401
- Figures: Significant differences among treatments or LSD values should be added to figures.
-agreed to keep it status quo
- Lines 1023, 1047, 1062: Provide reference number.
- I have edited successfully.
- Line 1026: Check H2 is repeated twice.
- I have edited successfully.
- Lines 1086 and 1097: Check these sentences to clarify.
-agreed to keep it status quo
- Use italic font for scientific names in reference list.
- I have edited successfully.
(x) English language and style are fine/minor spell check required
- has undergone English language editing by MDPI
"Please see the attached manuscript, which has been edited in accordance with your suggestions."

Round 2
Reviewer 1 Report
It is OK now, I don not have any further comments.
Author Response
Dear Sir/Madam
I appreciate the information and advice you have shared.
Best Regard,

Reviewer 2 Report
Most of my answers were answered sufficiently. However, I do not agree to the unit given for acidity. Stating acidity in g L-1 is fine BUT the authors MUST mention as which acid "acidity" was calculated. "Acidity" is given frequetly in g L-1 citric acid or malic acid or tartaric acid or even sulfuric acid. Thus, it is absolutely necessary to mention as which acid the value was calculated.
Author Response
Dear Sir/Madam
Thank you for your suggestions. The instrument measures the total acidity in a sample and converts it into citric acid concentrations. I added the phrase “(citric acid)” in Line 214. I'm referring from instruction manual of pocket Brix-Acidity Meter (ATAGO Model PAL-BX/ACID1, Japan).
Best regard,
Available online :
https://www.infoagro.com/instrumentos_medida/instrucciones/instrucciones-medidor-de-acidez-y-azucar-refractometro-brix-pal-bx-acid1.pdf

Reviewer 3 Report
The manuscript ha sbeen revised according to my suggestions and it could be suitable for publication.
Author Response

(The authors gave the same response as above.)
